# Early Left Ventricular Diastolic Dysfunction, Reduced Baroreflex Sensitivity, and Cardiac Autonomic Imbalance in Anabolic–Androgenic Steroid Users

**DOI:** 10.3390/ijerph18136974

**Published:** 2021-06-29

**Authors:** Evangelia Joseph Kouidi, Antonia Kaltsatou, Maria Apostolos Anifanti, Asterios Pantazis Deligiannis

**Affiliations:** Sports Medicine Laboratory, Department of Physical Education and Sport Sciences, Aristotle University of Thessaloniki, 57001 Thessaloniki, Greece; kouidi@phed.auth.gr (E.J.K.); akaltsat@gmail.com (A.K.); manyfant@phed.auth.gr (M.A.A.)

**Keywords:** anabolic–androgenic steroids, athletes, baroreflex sensitivity, cardiac autonomic nervous system, cardiac function

## Abstract

The effects of androgen anabolic steroids (AAS) use on athletes’ cardiac autonomic activity in terms of baroreflex sensitivity (BRS), and heart rate variability (HRV) have not yet been adequately studied. Furthermore, there is no information to describe the possible relationship between the structural and functional cardiac remodeling and the cardiac autonomic nervous system changes caused by AAS abuse. Thus, we aimed to study the effects of long-term AAS abuse on cardiac autonomic efficacy and cardiac adaptations in strength-trained athletes. In total, 80 strength-trained athletes (weightlifters and bodybuilders) participated in the study. Notably, 40 of them using AAS according to their state formed group A, 40 nonuser strength-trained athletes comprised group B, and 40 healthy nonathletes (group C) were used as controls. All subjects underwent a head-up tilt test using the 30 min protocol to evaluate the baroreflex sensitivity and short HRV modulation. Furthermore, all athletes undertook standard echocardiography, a cardiac tissue Doppler imaging (TDI) study, and a maximal spiroergometric test on a treadmill to estimate their maximum oxygen consumption (VO_2_max). The tilt test results showed that group A presented a significantly lower BRS and baroreflex effectiveness index than group B by 13.8% and 10.7%, respectively (*p* < 0.05). Regarding short-term HRV analysis, a significant increase was observed in sympathetic activity in AAS users. Moreover, athletes of group A showed increased left ventricular (LV) mass index (LVMI) by 8.9% (*p* < 0.05), compared to group B. However, no difference was found in LV ejection fraction between the groups. TDI measurements indicated that AAS users had decreased septal and lateral peak E’ by 38.0% (*p* < 0.05) and 32.1% (*p* < 0.05), respectively, and increased E/E’ by 32.0% (*p* < 0.05), compared to group B. This LV diastolic function alteration was correlated with the year of AAS abuse. A significant correlation was established between BRS depression and LV diastolic impairment in AAS users. Cardiopulmonary test results showed that AAS users had significantly higher time to exhaustion by 11.0 % (*p* < 0.05) and VO_2_max by 15.1% (*p* < 0.05), compared to controls. A significant correlation was found between VO_2_max and LVMI in AAS users. The results of the present study indicated that long-term AAS use in strength-trained athletes led to altered cardiovascular autonomic modulations, which were associated with indices of early LV diastolic dysfunction.

## 1. Introduction

There are several references in the literature regarding the use of anabolic–androgenic steroids (AAS) as doping substances in sports and particularly their health effects, there are several references in the literature. Most studies on their morphological and functional adverse effects on various functional systems after acute or long-term administration are based on experimental animal results [1]. For ethical, legal, and methodological reasons, the reports are not very well documented in athletes [1,2]. A limitation of almost all similar studies regarding the cardiovascular side-effects of AAS in athletes is the fact that the results are based on cross-sectional studies with different subject populations rather than longitudinal data. The longitudinal studies using AAS in humans would give rise to ethical problems. From the few clinical studies and case reports, the most destructive effects of AAS concern the cardiovascular system. Their administration in significant and long-term doses favors the manifestation of atherosclerosis of blood vessels, direct toxic action on myocardial cells, myocardial fibrosis, destruction of the endothelium, and dysfunction of the autonomic nervous system [2,3,4]. Clinical manifestations of these disorders are the occurrence of cardiomyopathy, myocardial infarction, arterial hypertension, arrhythmias, and sudden cardiac death [5,6,7]. Especially for the effects of AAS on athletes’ cardiac autonomic nervous system, studies are minimal, and the results are controversial. It is argued that their administration increases the action of the sympathetic nervous system and reduces the vagal tone [8,9]. Thus, an imbalance of the two limbs of the autonomic nervous system occurs, which is proven by the alterations of the heart rate variability (HRV) indices. In addition, based on experimental research and recent studies in AAS users, there are indications of the decreased sensitivity of the peripheral baroreflex and the sensitivity of the Bezold–Jarisch reflex control of heart rate and blood pressure [10,11,12,13]. There is still no research that studies whether cardiac autonomic dysfunction and reduced baroreflex sensitivity (BRS) after long-term AAS abuse are independent or associated with morphological and functional cardiac remodeling. The present study aimed to investigate the function of resting peripheral BRS and HRV modulation in strength-trained athletes receiving AAS and correlate the findings with the left ventricular anatomical and functional indices, as well as their aerobic capacity levels.

## 2. Materials and Methods

There was an open call in the local training centers for strength-trained athletes, such as bodybuilders and weightlifters, to participate in the study, where the aim and the method of the study were clearly stated. Healthy males aged 18 to 45 years, with at least five years’ experience in weight training, were eligible participants. Exclusion criteria were smoking, alcohol use or other drugs use besides AAS, presence of any chronic disease, atrial fibrillation, and medically prescribed testosterone therapy. Information about medical history, exercise training regime, and AAS usage was obtained from all volunteers. Based on self-reported history, athletes were allocated either to current AAS users or athletes without a history of AAS use. All AAS users reported that they were using only oral and injectable AAS substances for at least the last three years. In total, 40 strength-trained male athletes using AAS for at least 3 years (group A) and 40 strength-trained athletes, nonusers (group B) participated in this cross-sectional study. Moreover, 40 age-matched healthy nonsmokers and nonathletes, who did not use any medication were served as controls (group C).

All volunteers were examined in the Laboratory of Sports Medicine of the Aristotle University of Thessaloniki, in Greece, an authorized sports cardiology center in Greece. All tests were conducted in the morning and interpreted by the same cardiologist blinded to the identity of the participants. They were asked to refrain from exercise and all dietary sources of caffeine and alcohol 24 h before their examination. The evaluation included clinical history, clinical examination, resting electrocardiogram, an echocardiographic study, a head-up tilt test using a 30 min protocol to evaluate baroreflex sensitivity and short-term heart rate variability (HRV), and a maximum cardiopulmonary exercise testing on a treadmill.

All participants gave written informed consent. The study was conducted under the Declaration of Helsinki. The Ethics Committee of the Aristotle University of Thessaloniki approved the study protocol (Approval Number EC-65321/2012, Thessaloniki, 16 July 2012).

### 2.1. Measurements

#### 2.1.1. Echocardiographic Study

Transthoracic echocardiography was performed using Vivid S70 (GE Medical, Horten, Norway) with an M5S phased-array transducer. All echocardiographic images were obtained and stored by an experienced cardiologist–ultrasonographer blinded to the identity of the participants. The studies were analyzed offline by two cardiologists using the Echopac version 201 (GE, Horten, Norway).

Measurements of the left ventricle (LV) and its walls were performed in the parasternal long-axis view by M-mode approach according to the American Society of Echocardiography guidelines [14]. LV mass was estimated from parasternal views using the Devereux formula: 0.8{1.04[([LVEDD + IVSd + PWd]^3^ − LVEDD^3^)]} + 0.6, where LVEDD, IVSd, and PWd represent LV end-diastolic diameter, interventricular septal, and posterior wall thickness in diastole, and RWT was calculated with the formula: (2x posterior wall thickness)/(LV internal diameter at end-diastole). LVMI was corrected for body surface area (BSA). LV ejection fraction (LVEF) and LA volume were estimated using the biplane method of disks. LA maximal volume (LAVi) was measured at the end-systole and corrected for BSA. LV diastolic function was assessed according to the American Society of Echocardiography and the European Association of Cardiovascular Imaging guidelines [15]. Pulsed-wave (PW) Doppler was performed in the apical four-chamber view to obtain mitral inflow velocities. E-wave and A-wave peak velocities and their ratio E/A was measured.

PW tissue Doppler imaging (TDI) was performed in the apical four-chamber view to acquire mitral annular velocities at the septal and lateral wall and measure early diastolic peak E’ velocity and late diastolic A’ velocity in both walls. E/E’ average ratio was obtained averaging the e’ velocity from the septum and lateral sides of the mitral annulus and was used to estimate LV filling pressures. TR systolic jet velocity was obtained with CW Doppler from parasternal and apical four-chamber view with color flow imaging to obtain the highest Doppler velocity.

#### 2.1.2. Arterial Baroreflex Sensitivity and Heart Rate Variability Assessments

Baroreflex sensitivity was assessed by the Task Force Monitor 3040i device (CNSystem, Graz, Austria). After lying in a supine position for 5 min, all participants were placed in a 60° head-up position for 30 min. During each test, the RR intervals (RRI) were assessed from a continuous electrocardiogram, while continuous arterial Blood pressure (BP) was obtained using photoplethysmography on the middle finger. Baroreflex sensitivity (BRS) was assessed by spectral analysis of systolic BP (SBP) and RRI changes and was estimated using the average regression of the baroreflex slope of the SBP/RRI relationship. Moreover, the baroreflex effectiveness index (BEI), which indicates the ramps in RRI and SBP, was estimated. The ramp count and event count were also estimated. The ramp count indicates at least 3 consecutive beats, where the SBP rose or fell, while the event count indicates the number of baroreceptor sequences, where for at least 3 consecutive beats, there is a rise or fall of SBP with a subsequent shortening or lengthening of RRI.

By the same device, power spectral analysis of the short-term heart rate variability (HRV) was obtained for assessing cardiac autonomic activity. The low frequency (0.04–0.15 Hz) spectral component of the R–R interval using normalized units (LFnu–RRI) was estimated as a marker of sympathetic activity. On the other hand, the high frequency (0.15–0.4 Hz) spectral component of the R–R interval (HFnu–RRI) was estimated as a marker of cardiac vagal activity. Finally, their ratio (LF/HF ratio) was estimated as a marker of sympathovagal balance.

#### 2.1.3. Cardiopulmonary Exercise Testing

Finally, each participant underwent a maximal cardiopulmonary exercise testing on a Trackmaster treadmill (Full Vision Inc, Newton, KS, USA) using a Bruce protocol. There was a continuous electrocardiogram, while BP was measured at the end of each 3 min stage. Expiration gases were analyzed using Med Graphics Breeze Suite CPX Ultima spiroergometric device (Medical Graphics Corp, Saint Paul, MN, USA). Maximum oxygen consumption (VO_2_max) was defined as the highest oxygen consumption obtained in the final 30 **s** of the test, characterized by a plateau of oxygen uptake despite further increases in work rate (steady time). The respiratory exchange ratio was higher than 1.10 in all tests. Measurements at maximum exercise included SBP and diastolic blood pressure (DBP), heart rate (HR), pulmonary ventilation (VE), and total exercise time (ExTime).

### 2.2. Statistical Analysis

Continuous variables were expressed as mean ± standard deviation. The Kolmogorov–Smirnov test was used to test the normality, a condition fulfilled by the data analyzed. Changes of variables within the groups were evaluated by one-way analysis of variance, with a group being the independent variable. Correlation coefficients were calculated according to Pearson analysis. All statistical analyses were performed using the Statistical Package for Social Sciences (SPSS, Chicago, IL, USA), version 20.0 software for Windows. The significance level was *p* < 0.05.

## 3. Results

The physical characteristics of the participants are presented in Table 1. There were no statistically significant differences in all variables among the three groups.

Results obtained from the cardiopulmonary exercise testing are presented in Table 2. All tests were terminated due to volitional exhaustion. The SBP at rest was found in group A to be increased by 8.8% (*p* < 0.05) and 7.4% (*p* < 0.05), in comparison with groups B and C, respectively. In group A, SBPmax was higher by 6.8% (*p* < 0.05) and by 8.8% (*p* < 0.05), compared to groups B and C, respectively. Additionally, group A had increased ExTime by 11.0% and VO_2_max by 15.1% (*p* < 0.05), compared to group B, and by 17.5% (*p* < 0.05) and by 18.3% (*p* < 0.05), compared to C, respectively.

Table 3 presents the results of the baroreflex sensitivity and HRV assessments. There were no statistically significant differences in ramp count, event count and HFnu–RRI among the three groups. The BRS and BEI indices in group A were found to be decreased by 13.8% (*p <* 0.05) and 10.7% (*p <* 0.05), compared to B, and by 16.1% (*p <* 0.05) and 6.4% (*p <* 0.05), compared to C, respectively. Moreover, LFnu–RRI and LF/HF ratio were increased in AAS users by 24.2% (*p <* 0.05) and 25.5% (*p <* 0.05), compared to group B, and by 27.6% (*p <* 0.05) and 46.8% (*p <* 0.05), compared to group C, respectively.

Results obtained from the echocardiographic study are listed in Table 4. Group A demonstrated a significant increase in LVM and LVMI by 9.3% (*p <* 0.05) and 8.9 % (*p <* 0.05), compared to group B, and by 38.4% (*p <* 0.05) and by 39.1% (*p <* 0.05), compared to group C, respectively. Moreover, group A had increased RWT by 16.2% (*p <* 0.05), LAVi by 22.3% (*p <* 0.05) and TR peak velocity by 44.4% (*p <* 0.05), compared to group C. Finally, in group A septal E’ and lateral E’ were by 38.0% (*p <* 0.05) and 32.1% (*p <* 0.05) decreased, compared to group B, and by 44.1% (*p <* 0.05) and 35.0% (*p <* 0.05), compared to group C, respectively. On the other hand, group A demonstrated a significant increase in E/E’ aver by 32.0% (*p <* 0.05), compared to group B, and by 60.4% (*p <* 0.05), compared to group C (Figure 1).

In group A, significant correlations were obtained between (a) the years of AAS use and E/E’ aver (r = 0.609, *p* = 0.001); (b) BRS and E/E’ aver (r = −0.426, *p* = 0.006) as well as Lateral E’ (r = 0.325, *p* = 0.041); (c) VO_2_max and LVMI (r = 0.372, *p* = 0.018).

## 4. Discussion

The use of AAS by athletes has been associated with cardiovascular disorders, either acute (such as arrhythmogenic SCD, thromboembolic episodes, or myocardial infarction) or chronic (such as hypertension, atherosclerosis, or LV hypertrophy and dysfunction) [1,2,3,4,5,6,7]. Our results indicate that systemic use of AAS in strength-trained athletes leads to altered cardiac autonomic and hemodynamic function when assessing spontaneous BRS and short-term HRV indices. Moreover, AAS use seems to enhance LV hypertrophy and may accelerate LV diastolic dysfunction, depending on the intake years. An association between indices of early diastolic dysfunction and BRS depression was found in our anabolic users. Additionally, an increased maximal cardiopulmonary efficiency was established in AAS users; this finding was associated with an increased LVMI.

The BRS indicates the function of the baroreflex arch and is strictly linked with heart rate and blood pressure fluctuations. Measurement of BRS respects the interbeat interval in milliseconds per unit change, known as HRV, and blood pressure in mm Hg. The increase or decrease in HRV in response to a reduction or elevation of BP by baroreflex may occur by activating either sympathetic or parasympathetic limb, or both [16]. BRS follows the synergy between the vascular and autonomic functions to guide the BP fluctuations within normal levels. Thus, BRS modulates blood pressure fluctuations by changing the HR, myocardial contractility, and peripheral resistance [17].

Exercise training can improve BRS in healthy individuals and patients with cardiac autonomic disorders [18,19]. Subramanian et al. [16] supported that athletic training positively influences baroreflex and autonomic function. Short-term exercise training lowered standing HR in postural orthostatic tachycardia syndrome, attributable to a training-induced increase in BRS [20]. Reduced BRS was shown to be associated with high blood pressure, whereas resetting the baroreceptor working range to a higher level was observed in hypertension [21,22]. Additionally, impaired baroreflex sensitivity was found in depressive disorders; such abnormality may be a predisposing factor for sudden death in patients with underlying cardiac disease [23]. Reduced BRS was also associated with obesity, diabetes, and metabolic syndrome in adolescents and adults and was an acknowledged cardiovascular risk factor [24,25,26]. The long-time appearance of increased BP creates a “resetting” of the baroreflex so that the BP responses to exercise are regulated around a higher defined point [27]. In the early stages of BRS dysfunction, subjects may have normal resting BP, and the abnormal pressure responses may be disclosed during effort [28]. In hypertensive patients with LV hypertrophy, the LV diastolic function has independent associations with BRS parameters obtained at rest [29]. Several studies in heart failure patients showed that sympathetic hyperactivity is triggered by lower BRS [30,31]. In heart failure patients, the increased plasma levels of angiotensin II, due to the activation of the renin–angiotensin system, cause alteration on baroreflex control of sympathetic activity and HR directly in the vasomotor and cardiac centers in the brain and the peripheral nerve terminals, facilitating norepinephrine free and inhibiting acetylcholine release [30,31,32,33,34,35]. These patients may appear with abnormalities of the sinus node.

In our study, the tilt test results showed that AAS users presented significantly lower BRS and BEI compared to nonusers by 13.8% and 10.7%, respectively. Recently, there was similar evidence for lower BRS and sympathovagal imbalance in AAS users [13]. Moreover, a correlation between BP and spontaneous BRS and arterial stiffness was observed. Beutel et al. [10] reported the appearance of hypertension with differential hemodynamic changes and alterations in the reflex control in HR after chronic stanozolol administration in rats. Testosterone-treated animals presented rest bradycardia, cardiac hypertrophy, alterations in baroreflex activity, and enhanced response to sodium nitroprusside [12]. It was suggested that chronic administration of either testosterone or cocaine elicits functional changes in the activity of brain neurons regulating baroreflex responses [10,12]. Additionally, direct effects of both drugs on the heart may mediate the baroreflex activity; chronic cocaine administration increases chronotropic actions of catecholamines, while testosterone inhibits noradrenaline reuptake from the heart, increases the levels of the pore-forming subunits, and in addition, the activity of T-type calcium channel causing reduced reflex bradycardia [12]. Although animal studies do not necessarily apply to humans, similar mechanisms could, in future studies, justify the effects of AAS on BRS in athletes. Moreover, long-term treatment with AAS in rats reduced the sensitivity of the Bezold–Jarisch reflex control of bradycardia and BP, possibly due to cardiac hypertrophy [11].

Sedentary individuals exhibit raised sympathetic tone even at rest and higher reactivity to any stress [32,33]. Subramanian et al. [16] observed that HRV (total power and SDNN) was higher in athletes. The parasympathetic tone was higher in terms of higher RMSSD and higher HF power. We reported similar cardiac autonomic adaptations in our previous studies in athletes and patients with chronic diseases following exercise training [34,35]. In the present study, there were decreased short-term HRV indices during the head-up tilt test in the strength training athletes using AAS, compared to nonuser athletes, representing a shift towards sympathetic modulation predominance. Similarly, chronic abuse of high doses of AAS in bodybuilders led to cardiac autonomic dysfunction [8]. Dos Santos et al. [13] supported a relationship between AAS use in athletes and imbalance in autonomic control of both the periphery and the heart. In experimental studies, chronic high-dose AAS administration in rats caused impairment of parasympathetic cardiac modulation, decreased HF power, and HRV [9,10,36]. It was supported that cardiac autonomic dysfunction caused by AAS use may induce arrhythmias and sudden cardiac death [10].

Regular endurance, resistance, and combined training improve BP levels in hypertensive patients [37]. The reduction of high BP with exercise training is mainly due to attenuation in peripheral vascular resistance, caused by a reduction in sympathetic nerve activity and an increase in arterial lumen diameters [38]. The relationship between cardiac sympathetic overactivity and its association with cardiovascular diseases, such as hypertension and heart failure, is well established [39]. Controversies exist concerning the effects of AAS on BP. Some investigators have observed increased BP in strength trainees using AAS, whereas others have not [1,2,5,6]. Neto et al. [8] supported that the high BP at rest was associated with increased sympathetic modulation and enhanced cardiac hypertrophy in bodybuilders using AAS [8]. In our study, a significant increase was found in SBP at rest and at maximum effort in AAS users, compared to nonusers and controls. A similar increase in SBP at rest and during maximal workload in athletes using AAS, compared to nonusers, was observed by D’Andrea et al. [40]. They suggested that AAS can also cause sodium and water retention, with a consequent increase in blood volume and pressure.

It is well known that exercise training causes cardiac structural and functional adaptations. The type (concentric or eccentric or mixed hypertrophy) depends on the type of exercise training (aerobic, strength, or mixed). Strength-trained athletes often demonstrate concentric type LV hypertrophy. That hypertrophy is benign, as associated with normal systolic and diastolic properties. The present study showed increased relative wall thickness and LV mass index in AAS users, compared to nonuser athletes. This reinforces that AAS enhances the LV hypertrophy observed in the strength-trained athletes as a cardiac adaptation to training. In similar studies, significant increases in posterior and septal wall thickness, LVM, and chamber diameter in AAS users, compared to nonuser strength-trained athletes, were reported [41,42,43]. Moreover, our results indicate that long-term AAS use may accelerate LV diastolic dysfunction, depending on the years of intake, which may be early identified with TDI use. Tissue Doppler imaging in the estimation of diastolic function in AAS users showed a significantly increased E/E’ average ratio and reduced early diastolic tissue velocities (septal E’ and lateral E’). Early subclinical impairment of both systolic and diastolic myocardial function, mightily associated with mean dosage and duration of AAS use, was noticed by D’Andrea et al. [40]. Notin et al. suggested that the decrease in LV relaxation properties might have been due to an alteration in the active properties of the myocardium since no wall thickening was obtained in AAS-using bodybuilders [44]. The impairment of LV function in long-term AAS users is an early indication of LV dysfunction and may be sufficient to increase the risk of heart failure [45]. An autopsy study of cardiac dimensions in 173 AAS users also demonstrated a significantly elevated cardiac mass [46]. This cardiac remodeling has similar characteristics to hypertrophic cardiomyopathy, showing a prevalence of cardiac fibrosis and impairment of systolic and diastolic LV function [46]. In an experimental study in rats, interstitial collagen increases, leading to loss of diastolic function [47]. It was suggested that high concentrations of AAS by activating cytoplasmic androgen receptors, cell membrane receptors, and secondary transmitters stimulate the renin–angiotensin–aldosterone system. Leading to an increased synthesis of myocardial fibers, LV hypertrophy, and hypertension [1]. In our study, no LV systolic function alteration was demonstrated in AAS users. On the contrary, Alizade et al. [48] reported that peak systolic right ventricle free wall strain and strain rate were reduced in bodybuilders using AAS. In former AAS users, impaired LV systolic function was also reported [49]. Other studies have reported no significant difference in cardiac dimensions and systolic and diastolic function between AAS users and nonuser weightlifters [50,51,52]. However, the assessment techniques play a certain role in the credibility of the findings. D’Andrea et al. supported that the strain rate imaging was a more sensitive technique that allowed more accurate evaluation of ventricular regional wall motion in AAS users than Doppler myocardial imaging and conventional ultrasound [40].

We demonstrated a correlation between early LV diastolic dysfunction indices and spontaneous BRS depression in athletes using AAS. The increased sympathetic activity in AAS users may be the link between diastolic dysfunction and BRS reduction. A similar correlation has not yet been described in the literature. An association between LV diastolic function and depressed BRS was demonstrated in different cardiovascular conditions, but the mechanism and causality were not established [53,54]. It was also reported that diastolic dysfunction evoked significant heart baroreflex impairment in hypertensive patients [29,55].

AAS abuse causes an increase in muscle mass, can speed up muscle recovery after intense training or injury, and allow users to train longer and harder. In an experimental study in rats, the administration of AAS increased the respiratory muscle mass and the diaphragm [56]. These effects on both skeletal and respiratory muscles may contribute to the improved performance observed in our AAS users. Thus, the cardiopulmonary test results showed that AAS users had significantly longer time to exhaustion by 11% and maximal oxygen uptake by 15.1%, compared to the nonuser athletes. Interestingly, aerobic capacity was positively correlated with LVMI. A few studies demonstrated that AAS increases endurance performance in athletes [51]. No significant correlations were found between cardiopulmonary testing indices and baroreflex sensitivity or HRV parameters in our athletes.

The results of our study should be interpreted in light of some limitations. The use of AAS was based on athletes’ reports, and the dose of AAS in each athlete was not quantified. We did not perform body composition measurements in the participants. Moreover, we did not measure the arterial stiffness involved in the whole mechanism of the BRS. Additionally, we did not use strain and strain rate imaging for the assessment of myocardial function. The parameters that could have influenced the autonomic activity, such as diet, stress, and environmental influence, were not measured. Finally, although we demonstrated a correlation between LV diastolic function indices and BRS in AAS users, we cannot support a cause–effect relationship.

## 5. Conclusions

Long-term AAS use in strength-trained athletes decreases BRS and short-term HRV. An improvement in aerobic capacity was found in AAS users, which was positively correlated with LVMI. An essential finding of the study is the correlation between early left ventricular diastolic dysfunction indices and reduced BRS in AAS users. The exact mechanism of this relationship should be explained in future, more extensive studies.

## Figures and Tables

**Figure 1 ijerph-18-06974-f001:**
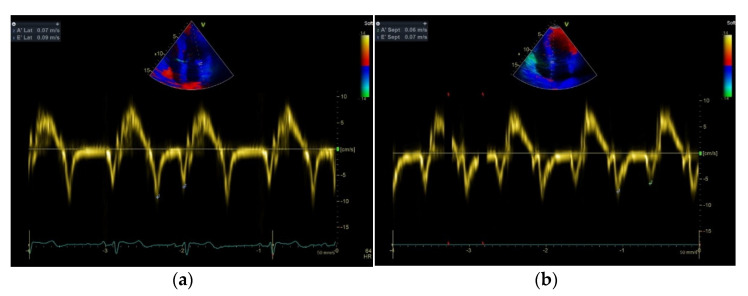
An example of tissue Doppler recordings of lateral (**a**) and septal (**b**) annular velocities from an athlete in group A.

**Table 1 ijerph-18-06974-t001:** Physical characteristics of the study population (mean ± S.D.).

Groups	A	B	C
**Age (years)**	27.4 ± 8.6	26.9 ± 7.8	27.1 ± 5.6
**Height (cm)**	1.75 ± 0.08	1.74 ± 0.08	1.76 ± 0.07
**Weight (kg)**	81.5 ± 15.2	81.3 ± 10.9	80.1 ± 8.1
**BMI**	26.6 ± 2.6	26.2 ± 2.3	25.9 ± 2.2
**Training age (years)**	10.3 ± 3.3	10.1 ± 3.6	-
**Duration of AAS use (years)**	4.3 ± 0.5		-

A: AAS users; B: nonusers; C: controls; BMI: body mass index; AAS: androgen–anabolic steroids.

**Table 2 ijerph-18-06974-t002:** Results from the cardiopulmonary exercise testing (mean ± S.D.).

Groups	A	B	C
**HRrest (beats/min)**	73.1 ± 12.1	72.5 ± 11.3	72.5 ± 9.3
**HRmax(beats/min)**	184.0 ± 11.3	184.7 ± 13.5	180.0 ± 11.7
**SBPrest (mmHg)**	127.0 ± 6.7 ^a,b^	116.7 ± 8.0	118.2 ± 9.2
**SBPmax (mmHg)**	174.0 ± 13.3 ^a,b^	162.9 ± 14.6	160.0 ± 12.3
**DBPrest (mmHg)**	75.7 ± 9.9	76.7 ± 9.1	77.2 ± 5.7
**DBPmax (mmHg)**	75.0 ± 7.8	75.0 ± 8.8	76.3 ± 6.6
**ExTime**	12.1 ± 1.1 ^a,b^	10.9 ± 1.2	10.3 ± 0.9
**VO_2_max (mL/kg/min)**	46.6 ± 6.4 ^a,b^	40.5 ± 7.1	39.4 ± 6.1
**VE max (L/min)**	107.7 ± 20.0	108.6 ± 19.4	109.1 ± 18.2

A: AAS users; B: nonusers; C: controls; ^a^ *p* < 0.05 A versus B; ^b^ *p* < 0.05 A versus C; HR: heart rate; SBP: systolic blood pressure; DBP: diastolic blood pressure; ExTime: exercise time; VO_2_ max: maximum oxygen consumption; VEmax: maximum ventilation.

**Table 3 ijerph-18-06974-t003:** Results of the Baroreflex sensitivity and HRV assessments (mean ± S.D.).

Groups	A	B	C
**BRS (ms/mmHg)**	9.4 ± 2.3 ^a,b^	10.9 ± 1.8	11.2 ± 1.9
**BEI (%)**	65.7 ± 10.4 ^a,b^	73.6 ± 9.2	70.2 ± 10.3
**Ramp Count**	333.6 ± 74.3	369.3 ± 74.9	355.5 ± 61.7
**Event Count**	172.1 ± 55.6	182.7 ± 45.5	176.5 ± 54.3
**HFnu-RRI (%)**	19.4 ± 4.7	20.5 ± 4.1	21.3 ± 4.3
**LFnu-RRI (%)**	97.5 ± 8.3 ^a,b^	78.5 ± 8.7	76.4 ± 8.8
**LF/HF ratio**	6.9 ± 3.9 ^a,b^	5.5 ± 3.6 ^c^	4.7 ± 3.7

A: AAS users; B: nonusers; C: controls; ^a^ *p* < 0.05 A versus B; ^b^ *p* < 0.05 A versus B; ^c^ *p <* 0.05 B versus C; BRS: baroreflex sensitivity; BEI: baroreflex effectiveness index; HFnu–RRI: high-frequency spectral component of the R–R interval using normalized units; LFnu–RRI: low-frequency spectral component of the R–R interval using normalized units; LF/HF ratio: low-frequency/high-frequency ratio.

**Table 4 ijerph-18-06974-t004:** Echocardiographic results (mean ± S.D.).

Groups	A	Β	C
**IVSd (mm)**	12.2 ± 1.6 ^b^	11.8 ± 1.1	10.6 ± 1.0
**PWd(mm)**	11.9 ± 1.4	11.5 ± 1.3	9.8 ± 1.3
**LVEDD (mm)**	51.8 ± 3.8	50.2 ± 4.3	49.9 ± 3.7
**LVM (g)**	227.0 ± 27.6 ^a,b^	208.7 ± 28.3 ^c^	164.0 ± 17.3
**LVMI(g/m^2^)**	115.2 ± 6.3 ^a,b^	105.8 ± 5.8 ^c^	82.8 ± 5.4
**RWT (cm)**	0.43 ± 0.05 ^b^	0.44 ± 0.05 ^c^	0.37 ± 0.04
**EF (%)**	62.7 ± 5.2	62.3 ± 5.0	62.9 ± 5.7
**LAVi (ml/m^2^)**	30.7 ± 1.8 ^b^	28.2 ± 2.1	25.1 ± 1.9
**TR peak velocity (m/s)**	1.3 ± 0.2 ^b^	1.1 ± 0.2	0.9 ± 0.2
**MVE (cm/s)**	73.8 ± 3.9	75.0 ± 3.2	72.1 ± 3.5
**MVA (cm/s)**	44.2 ± 2.1	45.1 ± 2.0	46.3 ± 1.9
**E/A**	1.65 ± 0.29	1.67 ± 0.28	1.52 ± 0.30
**E/E’ aver**	9.24 ± 1.42 ^a,b^	7.00 ± 0.9	5.76 ± 0.8
**Septal E’ velocity (cm/s)**	6.2 ± 0.70 ^a,b^	10.0 ± 0.9	11.1 ± 0.82
**Septal A’ velocity (cm/s)**	5.6 ± 0.6	6.7 ± 0.6	7 ± 0.7
**Lateral E’ velocity (cm/s)**	9.1 ± 0.6 ^a,b^	13.4 ± 0.5	14 ± 0.5
**Lateral A’ velocity (cm/s)**	7.1 ± 0.5	7.9 ± 0.6 ^c^	8 ± 0.7

A: AAS users; B: nonusers; C: controls; ^a^ *p* < 0.05 A versus B; ^b^ *p* < 0.05 A versus C; ^c^ *p <* 0.05 B versus C; IVSd: septal wall thickness in diastole; PWd: posterior wall thickness in diastole; LVEDD: LV end diastolic diameter; LV mass: left ventricular mass; LVMI: LV mass index; RWT: relative wall thickness; EF: ejection fraction; LAVi: LA maximal volume index; TR: tricuspid regurgitation systolic jet velocity; MVE: mitral peak E-wave velocity; MVA: mitral peak A-wave velocity; Septal E’ velocity: mitral annular early diastolic peak E –wave velocity in septum; Septal A’: mitral annular late diastolic peak A –wave velocity in septum; Lateral E’: mitral annular early diastolic peak E –wave velocity in the lateral wall; Lateral A’: mitral annular late diastolic peak A –wave velocity in the lateral wall; E/E’ aver = ratio of the early diastolic transmitral flow velocity to the average of septal and lateral early diastolic mitral annular velocity.

## Data Availability

The data presented in this study are available on request from the corresponding author. The data are not publicly available due to the privacy of included athletes.

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
