# Peer review of "Early Left Ventricular Diastolic Dysfunction, Reduced Baroreflex Sensitivity, and Cardiac Autonomic Imbalance in Anabolic–Androgenic Steroid Users"

_ijerph, 2021, doi:10.3390/ijerph18136974_

Round 1
Reviewer 1 Report
In this manuscript the authors sought to determine the function of resting peripheral BRS and HRV modulation in strength-trained athletes receiving anabolic-androgenic steroids as doping substances and correlate the findings with the left ventricular structural and functional indices, as well as their aerobic capacity levels.
I find the manuscript interesting, However some matters that have to be addressed before publication.
The authors make a bold statement in the first part of the introduction: “there is no information about cause- and-effect relationships between these effects and the cardiac structural and functional remodeling”. However, by just using the terms “Anabolic-Androgenic Steroid AND heart rate” in Pubmed, I found a very interesting review paper (DOI: 10.1016/j.amjcard.2010.05.013) already published in 2010 in which they reviewed a total of 49 studies describing 1,467 athletes, to investigate the cardiovascular effects of the abuse of AAS. This paper showing effects on left ventricular structural and functional indices, lipoprotein concentration, blood pressure and outcome was not refered to in the current manuscript. Why not? Furthermore, I could quickly find multiple papers on the Pubmed-term: “Anabolic-Androgenic Steroid Users AND Left Ventricular Diastolic Dysfunction” (Aaron L Baggish, et al. Circulation (2017), Barbosa Neto O, et al. Clin Auton Res (2018)).
All in all I have my doubts on the novelty of the current findings.
Nevertheless, I thinks that the experiments performed in the manuscript are well deigned and executed and that the provided results add to our understanding of the effects of long-term AAS usage.
Minor:
Line 121: 2x “All participants”
Reviewer 2 Report
Kouidi et al. describe the effects of anabolic-androgenic steriod usage on the baroreceptor reflex and cardiovascular function in strength-trained athletes. They report a reduced baroreceptor reflex response and dialstolic dysfunction that is correlated to years of AAS usage. This paper is written clearly and the results presented defend the conclusions drawn.
The authors describe a study using stanozolol in rats that described similar findings to those reported in this study. Can the authors differentiate between the effects of testosterone and cocain at a molecular level in those animals, and speculate a mechanism of action for their study based on the animal results?
Can the authors speculate why some studies see systolic dysfunction in AAS-users but other studies (including this one) see no effect on systolic function? Is it due to patient population differences? Assessment techniques?
There are a few instances where the use of an acronym is replaced with the complete word (ie: HF power and high-frequency power in lines 275 and 284). Once the acronym has been defined, stay consistent in using it. Other minor grammatical changes:
line 227: remove "indices"
line 234: remove comma after "may,"
line 257: should read "These patients may appear with abnormalities..."
Line 272: Should read "Sedentary individuals exhibit raised..."
Line 277: Should read "In the present study, there is decreased..."
Line 316: "mightily" should be replaced with "strongly"
Line 334: should read "...cardiac dimensions as well as systolic and..."
Round 2
Reviewer 1 Report
No further comments.